# Aging in rural communities: Engagement in indoor leisure activities and older adult health

**Pei-Yi Weng[1], Dongying Li[2], Man-Li Liao[3], Yen-Cheng Chiang[4]***

1 Department of Plant Industry, National Pingtung University of Science and Technology, Pingtung, Taiwan, R.O.C., 2 Department of Landscape Architecture & Urban Planning, Texas A&M University, College Station, Texas, United States of America, 3 Graduate Institute of Landscape Architecture and Recreation Management, National Pingtung University of Science and Technology, Pingtung, Taiwan, R.O.C., 4 Department of Landscape Architecture, National Chiayi University, Chiayi City, Taiwan, R.O.C.

* ycchiang@mail.ncyu.edu.tw

## Abstract

Aging is a pressing concern worldwide, particularly in rural communities characterized by a high aging index and an exodus of young individuals. Physical and mental well-being play key roles in older adults' overall health. COVID-19 has resulted in limitations on the outdoor activities of older adults, negatively affecting their social interactions and health. In this study, we designed an intervention to investigate the effects of indoor leisure activities on successful aging. Three types of activities were selected: horticultural, handicraft, and baking activities, each lasting 4 weeks. 82 older adults were randomly assigned to perform the activities and completed self-reported measures regarding their activities of daily living, depression, and mental and social health. Our results indicated horticultural activities to reduce depression and significantly improve physical, mental, and social health; handicraft activities likewise significantly improved physical, mental, and social health. Thus, indoor leisure activities can enhance the physical and mental health of older adults.

## Introduction

Since the second half of the 20th century, the global population has been aging. According to estimates by the United Nations, the global population of older adults is projected to exceed 425 million by 2050 [1,2]. This phenomenon of population aging is not limited to a single region, and the proportion of older adults continues to increase with increasing lifespans. Consequently, addressing the needs of older adults has emerged as a major global concern [3–5]. According to Tu et al. (2023) [6], the rate of population aging in Taiwan is particularly high; Taiwan is tied with Japan as the third-fastest aging country worldwide, following only South Korea and Singapore. In 1993, adults aged 65 and above comprised 7% of Taiwan's population, compared with 14% in 2018. Thus, the proportion of older adults in Taiwan doubled in just 25 years. For comparison, the corresponding doubling intervals in the United States and France were 73 and 115 years, respectively [7].

**Data availability statement:** All relevant data are within the paper and its Supporting information files.

**Funding:** This work was supported by the Agency of Rural Development & Soil and Water Conservation, MOA (former Bureau of Soil and Water Conservation, Council of Agriculture, Executive Yuan) of Taiwan. Grant Number: #110-1.1.1-1.1-020(3).

**Competing interests:** The authors have declared that no competing interests exist.

The challenges posed by population aging are particularly pronounced in rural communities because the rapid development of cities affords job opportunities to young individuals, causing them to relocate from rural areas. Consequently, rural areas undergo comparatively faster population aging [8]. Aging is associated with many problems, such as increased susceptibility to feelings of loneliness. According to Kumar et al. (2023) [9], approximately 17% of older adults who live in rural communities in India have depression. In addition, physiological function declines with age [10], and the diminishing living space of older adults has a detrimental effect on their physical and mental health.

In March 2020, the World Health Organization declared COVID-19 as a highly infectious and severe acute respiratory disease and classified it as a global pandemic [11]. Consequently, a series of policies and measures were implemented to safeguard public health, including public venue lockdowns, quarantine protocols, and social activity restrictions [12]. However, these policies had various unintended consequences, the brunt of which were borne by older adults [13]. For example, quarantine led to reduction in physical activity levels, which had adverse effects on physical and mental health [14,15], well-being [16], and social participation [17,18]. In a study conducted in the United Kingdom, Pierce et al. (2020) [19] discovered women to be more susceptible to mental health problems and to experience more severe mental health problems than men. Ultimately, implementing indoor leisure activities is essential to promote the physical and mental health of older adults during the pandemic.

## Depression among older adults living in rural communities

Because urban areas typically undergo rapid development and economic growth, rural communities experience reduced economic opportunities and an increasing exodus of young individuals, resulting in rural depopulation [20,21]. Consequently, the effect of population aging on rural communities surpasses the effect on urban communities [22]. In China, many older adults, referred to as empty nesters, tend to remain in rural communities [23,24]. These individuals face several challenges in performing their daily activities, and they exhibit lower self-care capabilities and experience reduced social engagement. Consequently, a higher proportion of older adults in rural Chinese communities as compared to urban areas have mental health problems such as depression and anxiety [25–27], and such problems tend to be magnified when coupled with conditions such as dementia and other chronic diseases [28–30]. More broadly, Huang et al. (2019) [31] discovered that the prevalence of depression among older adults in recent years has exceeded the rates observed in the 1990s and 2000s. Puffer and Miller (2001) [32] estimated that one out of every six older adults has depression, and Kumar et al. (2023) [9] estimated that approximately one-fifth (20.6%) of all older adults in India have depression. Overall, depression is one of the most common disorders among older adults [33,34]. Previous research on depression among older adults has revealed numerous contributing factors, including gender, socioeconomic status [28,35], physical health [14,15], social support [25], and social participation [36].

In Taiwan, many older adults live a "homebody" lifestyle characterized by prolonged periods spent indoors, inadequate physical activity, and feelings of loneliness and depression due to limited social activities [37]. This problem is particularly pronounced in rural communities [38], which often lack comprehensive medical services and other resources, a deficit that promotes the exodus of young individuals to urban cities. Consequently, as observed in China and elsewhere, older adults living in rural communities in Taiwan often face a lack of companionship [39] and experience more severe mental health problems compared with their urban counterparts [40,41].

With the advancement of medical knowledge and increasing emphasis on health awareness, many researchers have started to focus on the mental health of older adults [42]. According to multiple studies investigating various strategies aimed at mitigating depression, regular participation in social activities can alleviate depressive symptoms [43–45]. An appropriate level of physical exercise can also enhance mood and reduce depression severity. Hence, sufficient engagement in both social activities and physical exercise is closely linked to the alleviation of depression and has a substantial effect on both physical and mental health [36]. In this study, we assert that indoor leisure activities can improve social interactions between older adults and reduce their depression levels. Engagement in some leisure activities is associated with reduced depression in older adults, the study that found more frequent baking or cooking something special was only associated with lower subsequent depression in the older adults [46].

## Social health of older adults living in rural communities

Social health, also known as social adaptation, refers to an individual's ability to interact with others and with their social environment while maintaining positive interpersonal relationships and fulfilling their social role. As individuals grow older, the decline in their physiological function and self-care or daily living skills limits the time that they can allocate to social activities, thereby adversely affecting their social health. Social health is a fundamental component in addressing the challenges of population aging; older adults should be encouraged to alleviate their depression through increased social participation [43], that is, engagement in community activities that offer opportunities for interaction with others [44]. It is well-supported that social participation can enhance the physiological function and overall health of older adults [47] and also reduce their mortality rate [48]. Social participation is also regarded as a vital component in the physiological, mental, and emotional well-being of older adults [49], as active participation can effectively reduce risk of depression [43] and enhance life satisfaction and mental health [50]. Furthermore, even physical activities benefit from a social setting; according to Han et al. (2023) [51], group training yields more favorable outcomes than do individual activities.

Another factor influencing social participation is the place of residence. Rural communities typically offer a narrower array of activities compared with urban areas, leading older adults living in rural settings to exhibit a considerably lower level of social participation [52]. Rural dwellers are also more likely to experience anxiety and loneliness [36]. In the COVID-19 pandemic, various quarantine policies were implemented to control the spread of the virus, which had the collateral effects of considerably reducing social interactions and promoting feelings of isolation and loneliness among the public [53]. The effects on emotional and mental health varied among different demographic groups, with the greatest impacts being on minority groups, including patients with mental health problems and older adults [54]. For older adults in particular, this limited social environment led them to experience increased nervousness and anxiety, thereby increasing the risk of deterioration in their mental and social health [36,55] and of developing various diseases [56,57]. Among older women, extreme behaviors of suicide and self-harm became more prevalent.

Overall, social participation plays a key role in achieving successful aging, with neighborhood activities and interactions contributing to an improved quality of life and overall health [58]. In this study, we argue that social health correlates with the physical and mental health of older adults.

## Mental health of older adults living in rural communities

Individuals tend to prioritize their physical well-being while inadvertently neglecting their mental health. Nonetheless, as described by Godos et al. (2023) [59], mental health is the most critical component of successful aging. Aging is associated with declines in both physiological and cognitive functions, which in turn result in reduced independence, mobility, and quality of life [60]. Such impaired performance in activities of daily living is closely linked to physical health, often imposing limitations on the body and consequently affecting mental health [61]. Mental factors have a great effect on physical health, and older adults tend to be susceptible to mental health problems, indicating the interplay between mental factors and physiological function.

For older adults, poor mobility and factors such as social isolation and the pandemic reduce their likelihood of participating in social activities [53]. For instance, retirement may reduce opportunities to engage in social interactions, and the quarantine measures implemented for the pandemic may contribute to a sense of detachment from society [14]. This lack of social ties or connection leads older adults to experience negative emotions such as isolation and loneliness [62,63], which in severe cases may manifest as depressive tendencies [64,65]. Maintaining the health of older adults thus involves more than just disease prevention; it should also proactively address the enhancement of mental health and place emphasis on maintaining normal and balanced emotional, cognitive, and behavioral aspects to help older adults develop and maintain their sense of self-worth [53].

Participation in leisure activities of various types provides an array of physical, mental, and spiritual benefits. For older adults, long-term engagement in leisure activities represents a manifestation of successful aging, with the potential to enhance their mental health [66]. Consequently, continuous participation in such activities can allow older adults to improve their physical, mental, and spiritual health and alleviate their depression. There is a growing body of evidence indicating that engaging in participatory horticultural activities [67,68]; artistic endeavors and craft [69]; can offer various advantages, such as enhancing well-being, quality of life, and overall health for older adults.

## Study purpose

In this study, our primary goal was to investigate how different indoor leisure activities affected the physical, mental, and social health of older adults during the pandemic. Horticultural activities are mild and non-invasive, making it suitable for individuals of all backgrounds. Previous studies have shown that horticultural activities can improve the life satisfaction of individuals and optimize their quality of life [67,70]. Cultural, arts, and creative activities represent one domain of leisure activity that may be particularly beneficial for improve health in older adults [46,71]. Engagement in baking or cooking is associated with reductions in subsequent depression in older adults [46]. These leisure activities can be done in the home, many of which are inexpensive or free. Despite these activities being more accessible than receptive horticultural and art craft activities, there is less evidence on the associations between them and depression in older adults.

Three types of activities pertaining to rural communities and accessible to older adults were selected, namely horticultural activities, handicraft activities, and baking activities. Thus, the three primary goals of this study can be articulated as follows:

- To determine the extent to which engaging in horticultural activities significantly influences the physical, mental, and social health of older adults;

- To determine the extent to which engaging in handicraft activities significantly influences the physical, mental, and social health of older adults; and

- To determine the extent to which engaging in baking activities significantly influences the physical, mental, and social health of older adults.

 

## Methods

### Participants and activity types

The study areas included rural communities in Chiayi County and Pingtung County, which counties were selected because of their high aging indices: Chiayi County has the highest aging index and Pingtung County the fourth-highest among all counties in Taiwan. Moreover, according to the National Development Council (2020) [7], the proportions of older adults and the aging indices in these two counties exceed the national average (mean proportion of older adults in Taiwan: 15.29, mean aging index: 119.71). After careful consideration, we selected three specific rural communities within the counties, namely (1) Caipu Community in Budai Township, Chiayi County, (2) Tushi Community in Lioujiao Township, Chiayi County, and (3) Tungpian Community in Neipu Township, Pingtung County, in which older adults comprise 30.9%, 30.3%, and 26.3% of the population respectively [72]. Regarding sample size, we conducted an a priori statistical power analysis using G*Power 3.1.9.6 [73], which revealed that a minimum total sample size of 81 was required (with at least 27 samples for each group) to achieve a power of 0.95 for detecting a low effect size ($F = 0.20$) in an F test of repeated measures for within-between interactions [74]. The inclusion criteria for this study were as follows: (1) 60 years of age or older, (2) mentally conscious with no cognitive disorders, language barriers, or communication problems, (3) able to participate in the prescribed indoor activities and willing to complete all questionnaires, and (4) able to complete the Short Portable Mental Status Questionnaire in the pretest [59,75] with no more than two incorrect responses to ensure that mental state did not affect the research results. There was no restriction on the gender of participants. We did consider factors such as the development of the pandemic and the health of older adults over time, which may affect their ability to fully participate in our 4-week experiment and may also affect the statistical results of the research samples. In light of these considerations, 30 individuals were recruited for each type of activity, and each individual was instructed participate in only one activity. Hence, for this between subject study design, and we aimed to recruit a total of at least 90 participants for the three types of activities.

This study featured three distinct indoor leisure activities: horticultural activities, handicraft activities, and baking activities, each of which were implemented in one community: Tungpian Community, Caipu Community, and Tushi Community, respectively. To minimize travel inconvenience for participants, group assignments were made based on their respective community locations. Specifically, each community was designated to carry out only one type of activity: Tungpian Community residents participated in horticultural activities, Caipu Community residents engaged in handcraft activities, and Tushi Community residents took part in baking activities. Each activity was undertaken one session per week over a 4-week period [76], with each session lasting approximately 2 hours to ensure that activity duration did not influence the results. The details of each activity follow:

(1) *Horticultural activities*: Interacting with natural environments and engaging in horticultural activities can effectively regulate physical and mental health [77,78]. To ensure an effective intervention and operation during our horticultural activities, we enlisted researchers with a US horticultural therapy license to design and oversee the activities. Throughout the activities, the participants interacted with flowers and plants while providing mutual support. These interactions engaged their senses, improved their physical and mental health [79], and also influenced their social interactions and emotions [70]. During this 4-week program, four key topics were covered: orchid and wood art, combined potted planting, sowing and repotting, and *Tillandsia* planting. These topics aimed to enhance the participants' hand–eye coordination, hand joint flexibility, and communication skills. During the program, the participants observed the life cycle of plants, witnessing their transformation from nothingness to greenness, thus learning about the resilience of life. After each session, all plants were placed at the community center to encourage older adults to visit the center, observe and monitor others' plants, interact with others, and indirectly achieve a sense of self-identity.

(2) *Handicraft activities*: Previous studies have indicated that activities related to horticulture, woodwork, drawing, weaving, carving, and baking can empower individuals, alleviate pain, reduce anxiety, and instill a sense of safety and

serenity. When an individual relaxes, their heart rate, blood pressure, breathing frequency, and muscle tension tend to decrease [80,81]. Hence, we used various handicraft activities to offer diverse experiences to older adults and examined their effects on physical and mental health. During this 4-week program, the participants were asked to create two bamboo objects of varying complexity, a basket and a pendant. To help the participants become accustomed to the repetitive actions involved in bamboo weaving, the basket weaving activity was divided into three sessions. Professional bamboo weavers and volunteers provided skill instruction, enabling participants to enhance their hand–eye coordination and stimulate their cognitive functions during the weaving process. In the fourth week, participants learned how to create bamboo pendants, which are easier to craft. Ultimately, they completed most of the tasks independently, fostering a sense of self-identity and achievement.

(3) **Baking activities:** During this 4-week program, peanuts, being a local common crop, were centered as the primary ingredient. Professional bakers and volunteers devised recipes while incorporating the principles of a healthy diet. They selected ingredients with low sugar and salt content and employed actions such as filling, stacking, scooping, and spreading to enhance the flexibility of hand joints and hand–eye coordination. Participants were divided into separate groups during each activity to facilitate communication and interaction. After each session, the participants were allowed to take home the cookies or cakes that they had baked, thus enabling them to develop a sense of self-identity and achievement [82,83].

## Ethical approval statement

This study was conducted with approval from the Human Research Ethics Committee of National Cheng Kung University, Taiwan (HREC-E-110-125-2). The main purpose of the study was explained in writing, and all participants provided consent to participate by completing the survey.

## Measurements

The following measurement instruments were used in this study: the Instrumental Activities of Daily Living (IADL) scale, the Geriatric Depression Scale (GDS), a social health scale, and the Short Warwick–Edinburgh Mental Well-Being Scale (SWEMWBS).

**IADL scale.** We used the IADL scale [84], which is designed to gauge individual self-care skills, to evaluate participants' activities of daily living over the preceding month. These activities included shopping, housecleaning, managing finances, preparing meals, using transportation, using a telephone, doing laundry, and taking medication. The objective was to determine whether the participants were able to live independently in their own homes. Individuals who complete tasks alone or with minimal assistance are assigned a score of 1, and those who rely on others completely are assigned a score of 0. The total score ranges from 0 to 8, with a higher score indicating a stronger ability to perform activities of daily living. To ensure the reliability of the scale, we evaluated its internal consistency; this yielded a Cronbach's α of 0.92, with a test–retest reliability of 0.93. Previous studies support that the IADL scale reliably reflects physiological indices [85,86].

**GDS.** We used the simplified GDS, developed by Yesavage and Sheikh (1986) [87], to measure the self-perceptions of participants over the preceding week. The GDS consists of 15 self-reported items that evaluate the depressive tendencies of older adults, rather than diagnosing or describing depression as such. Specifically, the scale focuses on assessing emotional states such as sadness, apathy, and boredom. To minimize the potential fatigue and memory-related challenges that older adults may face when completing the scale, all responses are limited to a binary choice of "Yes" or "No." The total score of the GDS ranges from 0 to 15. A score of 0–4 indicates a normal emotional state, 5–10 mild depression, and 11–15 severe depression; thus, a higher score indicates more severe depression.

**Social health.** We drew upon the work of Rowe and Kahn (1997) [88] to examine the perspective of social health and successful aging. According to this perspective, robust social support is a critical factor in promoting the well-being of older

adults. Therefore, we asked our participants to reflect on the emotional support (e.g., respect or understanding) that they received over the preceding week from various sources, such as their family, friends, and others in their social circle. We also asked them to state whether they were willing to participate in social activities. The scale used for this assessment comprises four items scored on a 5-point Likert scale, with endpoints ranging from 1 (*strongly disagree*) to 5 (*strongly agree*). The total score ranges from 5 to 20, with higher scores indicating greater social health. The items of the scale have high internal consistency, as indicated by a Cronbach's α of 0.88.

**SWEMWBS.** The SWEMWBS is a subjective measure used to evaluate the eudemonic and hedonic aspects of well-being. Previous studies have confirmed its excellent reliability and validity [89,90]. The scale comprises seven items, to which participants respond depending on their experiences over the preceding 2 weeks. Responses are assigned scores ranging from 1 (*never*) to 5 (*always*). All responses are tallied, resulting in a total score ranging from 7 to 35, with higher scores indicating greater mental health.

## Procedure

We conducted the experiment between 1 August and 30 November 2021. Before the experiment, we provided all participants with a clear explanation of the study and asked them to provide informed consent. For participants that were illiterate, we read the questionnaire items for them and asked them to provide verbal responses.

Each participant was assigned to one type of activity. At the baseline phase of the experiment, the participants were asked to complete a pretest, which consisted of the four aforementioned questionnaires (IADL scale, GDS, social health scale, and SWEMWBS). Subsequently, they participated in their respective activities (horticultural, handicraft, or baking) for approximately 2 hours per week, for a total of 4 consecutive weeks. After each activity session, they underwent a posttest, which consisted of the GDS, social health scale, and SWEMWBS in weeks 1–3 and all four questionnaires at the end of the experiment (Fig 1).

## Statistical analysis

All statistical analyses were conducted using IBM SPSS Statistics version 22.0 (IBM, Armonk, NY, USA). Two types of analyses were performed. First, paired-samples Wilcoxon signed rank test were conducted to compare the effects of the aforementioned activities on the self-care skills of older adults. Second, repeated-measures analysis of variance (ANOVA) was conducted to determine whether the depression levels, social health, and mental health of the participants differed after their engagement in the activities. Horticultural, handicraft, and baking activities were regarded as the between-subject factors, and time (pretest: T0, posttest: T1–T4) was regarded as the within-subject factor. Additionally, to control Type I error inflation from multiple comparisons, pairwise estimated marginal means were adjusted using Bonferroni correction.

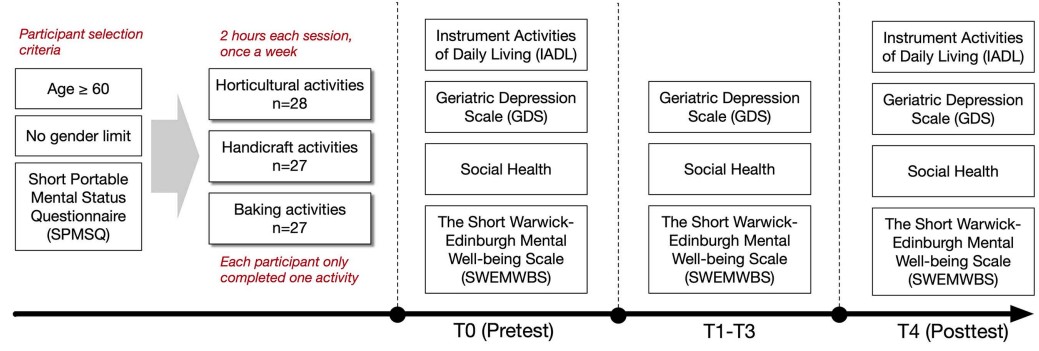

**Fig 1. Study procedure.**

## Results

### Participant characteristics

Data were collected during the peak of the pandemic, resulting in the absence of certain data due to factors such as participants requiring being required to make regular hospital visits or having health-related problems. A few participants dropped out, and the final number of participants in the horticultural, handicraft, and baking activity groups comprised 28, 27, and 27 individuals, respectively (Table 1).

Participants who engaged in the horticultural activities were on average 73.2 years old. Most were women (68.2%), and regarding education level, elementary and high school were tied for the most common category (40.9%). Those who engaged in handicraft activities had a mean age of 83.4 years (higher than in the other activities), were mostly women (73.3%) and overwhelmingly had received no formal education (86.7%; higher than in the other activities). Finally, participants who engaged in the baking activities averaged 75.8 years old, were mostly women (77.8%), and the greatest proportion had received elementary school education (38.9%). Overall, participants tended to be women, regardless of activity. Those who engaged in the horticultural activities were younger, those who engaged in the handicraft activities tended to be older, and those who engaged in the baking activities were more evenly distributed in terms of age. Most of the participants who engaged in the handicraft activities had not received formal education, whereas 95% of the participants who engaged in the horticultural activities had received formal education. As with age, those who engaged in the baking activities showed a more even distribution of educational level compared with the other groups.

### Description of activity effects on well-being

We evaluated the effects of indoor leisure activities, specifically horticultural activities, handicraft activities, and baking activities, on the physical and mental health of older adults. Each type of activity was undertaken for 4 weeks. Data on depression levels, social health, and mental health were collected over that period, with the participants being asked to complete four questionnaires as a pretest (T0) and posttest (T1 and T2). As shown in Table 2, depression levels among participants decreased as the number of sessions of horticultural and handicraft activities increased. In addition, the social and mental health of the participants improved as the number of sessions of horticultural activities increased.

### Effects of activities on the self-care skills of older adults

We took activities of daily living over the preceding month as a benchmark for evaluating participants' self-care skills; hence, we asked participants to complete a pretest in the first week and a posttest in the fourth week. A Wilcoxon signed rank test was conducted to examine the effects of the activities on self-care skills. The results indicated that both horticultural activities ($z = -3.559$, $p \leq .001$) and handicraft activities ($z = -1.939$, $p \leq .05$) significantly improved participants'

**Table 1. Descriptive characteristics of participants by activity type.**

| Activity (Community) | | Horticultural (Tungpian) n = 28 | | Handicraft (Caipu) n = 27 | | Baking (Tushi) n = 27 | |
|---|---|---|---|---|---|---|---|
| Variable | | n | % | n | % | n | % |
| Gender | Male | 9 | 31.8 | 7 | 26.7 | 6 | 22.2 |
| | Female | 19 | 68.2 | 20 | 73.3 | 21 | 77.8 |
| Education | None | 1 | 3.6 | 23 | 86.7 | 6 | 22.2 |
| | Elementary | 11 | 39.3 | 2 | 6.7 | 11 | 40.7 |
| | Junior high school | 5 | 17.9 | 2 | 6.7 | 5 | 18.5 |
| | High school | 11 | 39.3 | – | – | 5 | 18.5 |

**Table 2. Mean (SD) scores of depression, social health, and mental health at T0–T4 for the three activity groups.**

| | T0 | T1 | T2 | T3 | T4 |
|---|---|---|---|---|---|
| **Horticultural (n = 28)** | | | | | |
| Depression | 7.25 (2.01) | 4.75 (1.87) | 3.38 (2.18) | 2.42 (1.56) | 2.08 (1.14) |
| Social health | 13.88 (2.76) | 17.29 (2.71) | 17.33 (2.35) | 17.83 (2.37) | 18.13 (2.05) |
| Mental health | 23.91 (4.84) | 27.93 (4.67) | 29.24 (4.28) | 31.06 (3.65) | 31.49 (3.98) |
| **Handicraft (n = 27)** | | | | | |
| Depression | 6.39 (2.45) | 3.28 (1.27) | 2.89 (1.68) | 2.83 (1.20) | 2.22 (1.77) |
| Social health | 14.33 (1.72) | 17.56 (1.58) | 17.00 (1.72) | 16.89 (1.81) | 18.50 (1.29) |
| Mental health | 25.45 (5.67) | 26.93 (4.47) | 26.88 (4.23) | 28.61 (4.00) | 30.53 (4.55) |
| **Baking (n = 27)** | | | | | |
| Depression | 5.13 (1.41) | 2.53 (1.36) | 2.73 (1.03) | 1.80 (1.26) | 1.40 (1.30) |
| Social health | 15.00 (2.14) | 17.80 (1.57) | 17.73 (2.09) | 18.47 (3.00) | 18.27 (2.31) |
| Mental health | 24.17 (5.94) | 30.56 (5.33) | 29.24 (4.94) | 32.33 (3.62) | 33.10 (2.85) |

self-care skills. However, no significant difference was observed before and after the baking activities ($z = -1.811$, $p = .07$). In sum, after older adults participated in horticultural and handicraft activities, their ability to perform activities of daily living such as shopping, going outdoors, preparing meals, housecleaning, doing laundry, using a telephone, taking medication, and managing finances was considerably improved (Fig 2).

### Effects of activities on the depression level, social health, and mental health of older adults

We next investigated whether the selected activities affected participant depression level, social health, and mental health using repeated-measures ANOVA. Depression level was significantly affected by all three activities ($F = 4.95$, $p \leq .05$), with a significant difference of GDS pretest and posttest results within the same activity ($F = 101.49$, $p \leq .000$). Additionally, there was a significant effect of the interaction between time and activity type ($F = 3.03$, $p = .003$). In other words, the selected indoor leisure activities effectively reduced depression level in older adults (Table 3). To control for potential Type I error due to multiple comparisons, Bonferroni correction was applied to pairwise contrasts. The Bonferroni-adjusted results were highly consistent with the original findings, indicating that the intervention effects remained robust after correction. Social health was also impacted, with a significant difference between the pretest and posttest results ($F = 58.91$, $p \leq .001$). In other words, social health was improved after participation in these activities (Table 3). Likewise, mental health showed a significant difference between the pretest and posttest results ($F = 46.20$, $p \leq .000$), and an interaction effect of time (pretest and posttest) and activity type ($F = 2.56$, $p \leq .05$). Taken together, all three types of activities improved the mental health of older adults (Table 3).

## Discussion

### Effects of horticultural activities on the physical, mental, and social health of older adults

In this study, we investigated the effects of horticultural activities on the physical, mental, and social health of older adults. Our results indicated that after the horticultural activities, participants' self-care skills were considerably improved, and their level of depression was significantly decreased. This is consistent with previous studies, which reported horticultural activities to influence both physical health [91,92] and mental health, such as through alleviating depression [23], which in turn mitigates symptoms of dementia [93,94]. Plants also reduce depression levels among older adults and mitigate the risk of dementia [91,92]. Ultimately, green care activities can indirectly enhance interpersonal interactions, boost social engagement, stabilize emotions, and elevate mood.

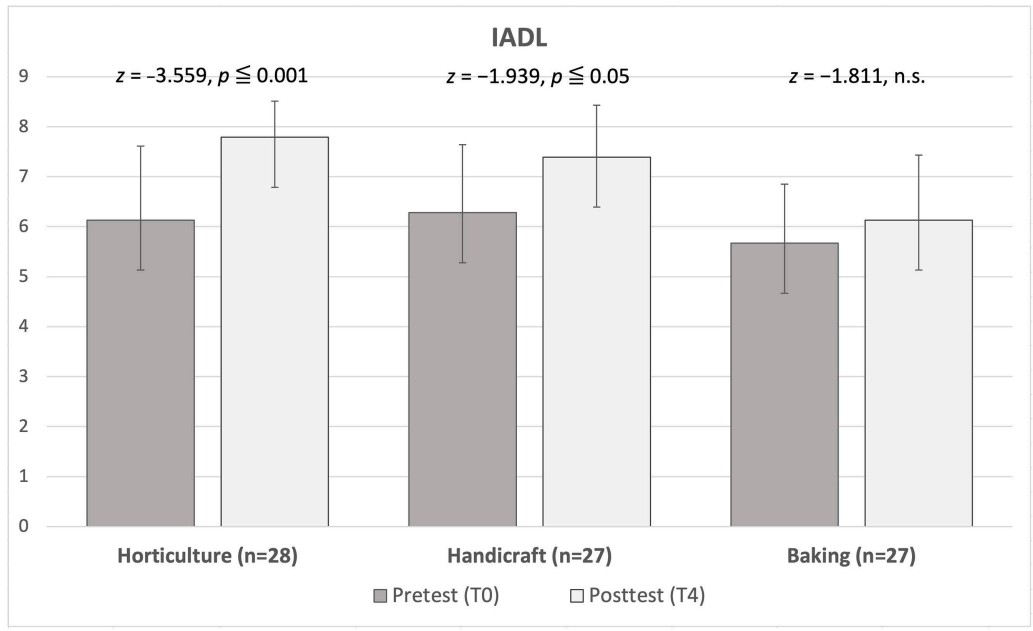

**Fig 2. Changes in Instrumental Activities of Daily Living (IADL) from T0 to T4 by activity group.** Values are mean±SD.

**Table 3. Repeated-measures ANOVA results for depression, social health, and mental health across time points.**

| Source | SS | df | MS | F | p | partial η2 |
|---|---|---|---|---|---|---|
| *Depression (GDS)* | | | | | | |
| Between subjects | | | | | | |
| Type of activity | 72.75 | 2 | 36.37 | 4.95 | 0.011 | 0.16 |
| Error | 396.50 | 81 | 7.34 | | | |
| Within subjects | | | | | | |
| Time | 641.78 | 4 | 160.45 | 101.49 | 0.000 | 0.65 |
| Time × type of activity | 38.30 | 76 | 4.79 | 3.03 | 0.003 | 0.10 |
| Error | 341.47 | 248 | 1.58 | | | |
| *Social health* | | | | | | |
| Between subjects | | | | | | |
| Type of activity | 18.48 | 2 | 9.24 | 0.61 | 0.545 | 0.02 |
| Error | 812.10 | 81 | 15.04 | | | |
| Within subjects | | | | | | |
| Time | 514.35 | 4 | 128.59 | 58.91 | 0.000 | 0.52 |
| Time × type of activity | 22.60 | 76 | 2.82 | 1.29 | 0.248 | 0.05 |
| Error | 471.50 | 248 | 2.18 | | | |
| *Mental health (SWEMWBS)* | | | | | | |
| Between subjects | | | | | | |
| Type of activity | 198.53 | 2 | 99.27 | 1.53 | 0.227 | 0.05 |
| Error | 3515.38 | 81 | 65.10 | | | |
| Within subjects | | | | | | |
| Time | 1679.68 | 4 | 419.92 | 46.20 | 0.000 | 0.46 |
| Time × type of activity | 186.12 | 76 | 23.27 | 2.56 | 0.011 | 0.19 |
| Error | 1963.46 | 248 | 9.09 | | | |

GDS assessments conducted throughout the study period revealed a significant difference over time, particularly after the third and fourth weeks. Specifically, longer participation in the horticultural activities correlated with lower depression levels. Overall, these findings are consistent with previous studies, which indicated that horticultural activities reduce negative emotions. Although horticultural activities appeared to reduce depression, this effect should be interpreted with caution, as pandemic-related stress may have influenced the outcomes. The findings suggest a supportive role for such activities, but stronger control designs are needed to confirm intervention-specific effects. It should be emphasized that the quasi-experimental before–after design during the COVID-19 pandemic did not fully meet the stability assumptions (e.g., environmental constancy, unaffected baselines). The rapidly changing pandemic environment may have confounded depression and social health outcomes. These limitations restrict causal inference, yet still provide valuable insight into how indoor leisure activities function under crisis conditions.

According to previous studies, individuals who engage in more social interactions tend to enjoy a more favorable physical and mental state. Horticultural activities are predominantly community-based activities that increase the likelihood of interaction between community members, thereby enhancing their relationships and improving their mental health [95], and boosting their social health [96]. However, our results indicated that horticultural activities did not have a significant effect on the social health of older adults. This conclusion contradicts those of previous studies. Notably, all activities in this study were conducted during the peak of the pandemic, a period during which people became more socially distant and older adults in particular tended to experience a decline in social health and develop negative emotions.

## Effects of handicraft activities on the physical, mental, and social health of older adults

We also investigated the effects of handicraft activities on the physical, mental, and social health of older adults. Our results indicated those who engaged in handicraft activities to show improved self-care skills, a positive effect that presumably attributable to their 4 weeks of engagement in weaving activities and interactions with other participants. Additionally, participants experienced improvements in their mental health in terms of alleviating depression, with significant differences after 1 week as well as after 3 and 4 weeks of handicraft activities. Participants who performed handicraft activities engaged in bamboo weaving during the first week, completed a bamboo candy basket in the third week, and almost independently crafted bamboo pendants during the fourth week. We speculate that the observed mental health improvement is at least partly attributable to the curiosity and sense of achievement that participants experienced while encountering novel tasks and successfully completing projects. Thus, engaging in handicraft activities that involve unique materials or techniques has the potential to stimulate older adults' curiosity, enhance their focus, and positively affect their mental well-being.

Previous studies have indicated that individuals may experience both physical and mental relaxation while creating clay sculptures [81,97]. Our results similarly indicated engagement in handicraft activities to have a positive effect on the mental health of older adults. As participants' mental health improved, they exhibited increased curiosity toward new challenges, which provided them with distraction and alleviated their suppressed emotions. Importantly, touch is the foundation for secure attachment. When individuals engage in handicraft activities, they use their sense of touch to receive stimulation and experience mental and spiritual healing [80]. Kramer (1971) [98] likewise argued that the process of creating art can provide mental and spiritual healing. Thus, engagement of older adults in handicraft activities can be empowering, reduce anxiety, instill a sense of security, enhance overall quality of life, and greatly improve health [99,100].

## Effects of baking activities on the physical, mental, and social health of older adults

Finally, we investigated the effects of baking activities on the physical, mental, and social health of older adults. Our results indicated that baking activities did not significantly impact the ability to perform activities of daily living. Given that most participants were women with comparable experience in baking and meal preparation, the baking activities did not confer any distinct benefits for participants' daily living abilities, despite the physical effort involved in baking. This finding

aligns with prior research indicating that the impact of physical activities on daily living performance is contingent upon the intensity, novelty, and relevance of the activity to the participants' daily lives [101]. Since the physical effort required for baking is relatively modest and similar to routine household tasks, it may not sufficiently challenge physical capacities to yield measurable improvements in activities of daily living performance. In contrast, structured physical activities targeting strength, balance, and coordination are more likely to enhance activities of daily living [102]. Likewise, baking activities were not found to have significant effects on the physical, social, and mental health of older adults, although the GDS posttest scores were lower than the pretest scores, indicating a trend toward reduced depression level after engaging in baking activities. The limited number of participants may have contributed to the lack of statistical significance for this reduction. As discussed by Haley and McKay (2004) [83], baking classes can boost confidence, enhance attention, and provide a sense of achievement. Increasing the sense of participation through specific activities can alleviate depression and promote behavioral activation, ultimately improving mental health [82].

Notably, a Belgian research team reported that their participants experienced improved emotions after consuming desserts, even when they were unable to taste, see the color of, or smell the desserts [103]. Renowned English dessert chef John Whaite shared his personal experience of a healing effect from baking, with the activity helping him navigate challenging times and ultimately curing his depression. Overall, our results concerning baking are not consistent with those of previous studies, presumably because the community in which our participants resided already had baking equipment available. Additionally, many government projects have implemented baking activities, resulting in a lack of novelty and curiosity among the participants, which contributed to a reduction in the physical and mental healing effects of baking. To address similar challenges in the future, activity designers should draw experience from previous activities and devise new baking techniques or dietary approaches that differ from those activities, the better to stimulate participant curiosity.

## Research suggestions and applications

Here we outline our suggestions for future research. First, stronger control over the environment of the activities and content of each week's program would be desired to rule out confounding factors. Second, data that offers further insights into participants' perceptions of the benefits of the activities and how various aspects of their health may be influenced can be collected. Especially with ecological momentary assessment and other biosensing technology, tracking mental health and moods during the intervention period becomes possible. This study focused only on older adults residing in rural communities, and hence featured a limited sample population. As the study was conducted during COVID-19, it offers unique insights into the effectiveness of such programs amidst public health crisis, but the results may not apply to other periods and business as usual times. Future studies should include more diverse groups of older adults, including those residing in urban areas and those belonging to different age groups.

Our research results can nonetheless serve as a valuable reference for designing activities for older adults that incorporate various learning approaches. Furthermore, activities can be designed to encourage older adults to engage in more indoor activities that benefit both their physical and mental well-being. These activities should be diverse to promote creative thinking. They could also incorporate local agricultural or cultural elements to tap into existing skills. The activities can not only foster a sense of achievement but also encourage increased social interactions with other community members, ultimately improving mental health. Our findings suggest that when activities are similar to those commonly performed by community members, they may become less engaging. Therefore, future studies on green care activities in older adults should consider designing unique baking or dietary courses and innovative, sensory-rich activities aimed at preventing development of disinterest, thereby maximizing the physical and mental benefits of the activities. Incorporating diverse activities such as stretching and floral arrangement can further enhance the participation of older adults.

Although this study employed a quasi-experimental design without a no-intervention control group, this approach was necessitated by the ethical and logistical constraints during the peak of the COVID-19 pandemic. In rural communities with limited access to mental health resources, withholding potentially beneficial interventions from a control group was

deemed inappropriate. Instead, the study utilized a between-subjects design with different communities assigned to distinct activity types, enabling comparative analysis. Repeated-measures ANOVA was employed to assess temporal trends, offering robust insights into within-group changes. Previous studies have indicated that repeated-measure designs, while lacking a traditional control, can still provide valid inferences under constrained conditions [104–106]. Nonetheless, future research should incorporate randomized controlled trials or delayed-intervention groups where feasible, to further strengthen causal inference and account for potential confounding factors such as seasonal effects or background health trends.

## Limitations

Limitations of this study need to be addressed. First, as participants were allocated to activity groups based on their community of residence rather than through random assignment, the potential for community-level confounding cannot be excluded. Consequently, the comparability between groups may be limited, and any causal interpretations of observed differences across activity types should be made with caution. Second, this study is the absence of a no-intervention control group, which restricts the ability to draw definitive causal conclusions about the effects of the leisure activities. While the use of a between-group design and repeated measurements provides some degree of control, the influence of extraneous variables such as seasonal mood variations or broader contextual factors cannot be entirely excluded. This design choice was made due to ethical and practical considerations during the pandemic, a time when social isolation and mental health risks were particularly high among older adults. Furthermore, the pandemic context likely introduced environmental changes that could not be fully controlled, which might have influenced mental health outcomes independently of the interventions. This limitation highlights the inherent challenges of before–after designs conducted in unstable environments, where external factors may confound intervention effects. Future studies are encouraged to adopt randomized or waitlist-controlled designs to enhance methodological rigor and isolate treatment effects more precisely.

## Conclusion

In this study, we examined the effects of three indoor leisure activities, namely horticultural, handicraft, and baking activities, on successful aging and on activities of daily living, depression levels, social health, and mental health in older adults during the COVID-19 pandemic. Our results indicated the three activities to each play a key role in enhancing physical and mental health. Both horticultural and handicraft activities emerged as effective means for improving the self-care skills of older adults and alleviating their depression. All three types of activities had a strong positive effect on depression levels and generally positive effects on social and mental health. Thus, indoor leisure activities have a discernible influence on the physical and mental health of older adults. Overall, our findings can serve as an objective and practical reference for shaping care policies for older adults. Governments should design and implement various activities tailored to older residents of rural communities to help them achieve successful aging and improve their overall health.

## Supporting information

**S1 File. Data 2021 PLos ONE.**
(CSV)

## Author contributions

**Conceptualization:** Yen-Cheng Chiang.

**Data curation:** Pei-Yi Weng, Man-Li Liao.

**Formal analysis:** Pei-Yi Weng.

**Funding acquisition:** Yen-Cheng Chiang.

**Investigation:** Pei-Yi Weng, Man-Li Liao.

**Methodology:** Pei-Yi Weng, Yen-Cheng Chiang.

**Supervision:** Yen-Cheng Chiang.

**Validation:** Pei-Yi Weng, Dongying Li, Yen-Cheng Chiang.

**Writing – original draft:** Pei-Yi Weng, Dongying Li, Yen-Cheng Chiang.

**Writing – review & editing:** Pei-Yi Weng, Dongying Li, Yen-Cheng Chiang.

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
