## [Decision Letter · Decision Letter 0]

16 Dec 2024

Aging in rural communities: Engagement in indoor leisure activities and older adult health

PLOS ONE

Dear Dr. Chiang,

Thank you for submitting your manuscript to PLOS ONE. After careful consideration, we feel that it has merit but does not fully meet PLOS ONE’s publication criteria as it currently stands. Therefore, we invite you to submit a revised version of the manuscript that addresses the points raised during the review process.

Please note that we have only been able to secure a single reviewer to assess your manuscript. We are issuing a decision on your manuscript at this point to prevent further delays in the evaluation of your manuscript. Please be aware that the editor who handles your revised manuscript might find it necessary to invite additional reviewers to assess this work once the revised manuscript is submitted. However, we will aim to proceed on the basis of this single review if possible. 

Could you please revise the manuscript to carefully address the concerns raised?

We look forward to receiving your revised manuscript.

Kind regards,

Helen Howard

Staff Editor

PLOS ONE

Journal Requirements:

“This work was supported by the Agency of Rural Development & Soil and Water Conservation, MOA (former Bureau of Soil and Water Conservation, Council of Agriculture, Executive Yuan) of Taiwan. Grant Number: #110-1.1.1-1.1-020(3).”

5. We note that you have indicated that there are restrictions to data sharing for this study. PLOS only allows data to be available upon request if there are legal or ethical restrictions on sharing data publicly. For more information on unacceptable data access restrictions, please see http://journals.plos.org/plosone/s/data-availability#loc-unacceptable-data-access-restrictions.

6. Please ensure that you include a title page within your main document. You should list all authors and all affiliations as per our author instructions and clearly indicate the corresponding author.

7. Your ethics statement should only appear in the Methods section of your manuscript. If your ethics statement is written in any section besides the Methods, please delete it from any other section.

Reviewers' comments:

Reviewer's Responses to Questions

**Comments to the Author**

1. Is the manuscript technically sound, and do the data support the conclusions?

Reviewer #1: Yes

2. Has the statistical analysis been performed appropriately and rigorously?

Reviewer #1: Yes

3. Have the authors made all data underlying the findings in their manuscript fully available?

Reviewer #1: Yes

4. Is the manuscript presented in an intelligible fashion and written in standard English?

Reviewer #1: Yes

Reviewer #1: Dear Autors,

I would like to express my gratitude for the opportunity to review the manuscript titled "Aging in rural communities: Engagement in indoor leisure activities and older adult health." In my view, this type of research, which examines the effectiveness of various physical activities in older adults, is highly relevant. From both a scientific and methodological perspective, I find the manuscript to be rigorous in its justification, design, analysis, and interpretation.

However, there are some areas that require clarification and improvement, which, if addressed, would enhance the overall quality of the manuscript:

• There is a lack of clarity regarding the selection and assignment of participants to each group. A more detailed explanation is needed regarding the criteria and process for categorizing participants into the respective groups (page 12, line 211);

• Additional clarification is required concerning the effects of physical activities on the ability to perform activities of daily living (page 24, lines 439-451)

**Do you want your identity to be public for this peer review?** For information about this choice, including consent withdrawal, please see our Privacy Policy

Reviewer #1: No

---

## [Author Response · Author response to Decision Letter 1]

21 Dec 2024

Please see the attached file point-by-point responses to reviewer.

---

## [Decision Letter · Decision Letter 1]

15 Jul 2025

Dear Dr. Chiang,

Thank you for submitting your manuscript to PLOS ONE. After careful consideration, we feel that it has merit but does not fully meet PLOS ONE’s publication criteria as it currently stands. Therefore, we invite you to submit a revised version of the manuscript that addresses the points raised during the review process.

I think the control group proposed by Reviewer 4 is worth discussing. Please provide a reasonable explanation or supplement relevant research by the author.

We look forward to receiving your revised manuscript.

Kind regards,

Bifeng Zhu

Academic Editor

PLOS ONE

Journal Requirements:

Reviewers' comments:

Reviewer's Responses to Questions

**Comments to the Author**

Reviewer #2: All comments have been addressed

Reviewer #3: All comments have been addressed

Reviewer #4: (No Response)

2. Is the manuscript technically sound, and do the data support the conclusions?

Reviewer #2: Yes

Reviewer #3: Yes

Reviewer #4: No

3. Has the statistical analysis been performed appropriately and rigorously?

Reviewer #2: Yes

Reviewer #3: Yes

Reviewer #4: No

4. Have the authors made all data underlying the findings in their manuscript fully available?

Reviewer #2: Yes

Reviewer #3: Yes

Reviewer #4: No

5. Is the manuscript presented in an intelligible fashion and written in standard English?

Reviewer #2: Yes

Reviewer #3: Yes

Reviewer #4: (No Response)

Reviewer #2: The revised manuscript is well-written, methodologically sound, and addresses a timely and socially relevant topic—the impact of indoor leisure activities on older adults' physical and mental health during the COVID-19 pandemic in rural Taiwan. The authors present a thoughtful design, appropriate measures, and a clear analysis using repeated-measures ANOVA.

The paper contributes to the literature on aging, leisure therapy, and rural health promotion. However, I recommend minor revisions before acceptance:

1. Clarify Group Assignment Strategy:While the authors explain that group assignment was based on location (to reduce participant burden), this introduces a risk of community-level confounding. Although randomization is not claimed, I suggest adding a brief statement in the Limitations section to acknowledge this limitation and its potential influence on between-group comparability.

2.Proofreading for Minor Language Corrections: A few grammatical or typographical edits are needed. For example:

Line 27: “complete self-reported their activities” → should be revised for clarity.

Line 449: “participants were women who were as familiar with baking as with meal preparation” → consider rephrasing for smoother readability.

Reviewer #3: The study has significant merit and high relevance for aging in rural communities. so it may be publishable. Paper has insights and it will help policy making in different stakeholder.

Reviewer #4: This manuscript addresses an important and increasingly relevant topic by exploring the impact of indoor leisure activities on successful aging in rural older adults. The authors present an intervention study comparing horticultural, handicraft, and baking activities and report positive effects on depression, physical, mental, and social health outcomes. The study’s focus on an aging rural population and its practical aim of promoting well-being in resource-limited settings are commendable.

I noted one major weakness in the study, which I believe makes the results unpublishable in their current form. The study uses a pre- and post-intervention design in the same participants, without any control group. This approach is highly vulnerable to confounding factors such as seasonal changes in mood or health. In the absence of control groups (e.g. no-intervention or delayed-intervention groups), the observed effects cannot confidently be attributed to the interventions.

There are also some minor weaknesses in the study, most notably the use of a paired t-test to compare IADL data. A Wilcoxon signed-rank test would be better, since the data would not be expected to be continuous, normally distributed, have equal variances between groups, and are also bounded at 0 and 8.

**Do you want your identity to be public for this peer review?** For information about this choice, including consent withdrawal, please see our Privacy Policy

Reviewer #2: No

Reviewer #3: No

Reviewer #4: No

---

## [Author Response · Author response to Decision Letter 2]

21 Jul 2025

We sincerely thank the reviewers for their detailed comments. The summary of our modifications to the manuscript for the reviewers’ concerns are on the attached response letter.

---

## [Decision Letter · Decision Letter 2]

5 Aug 2025

Dear Dr. Chiang,

Thank you for submitting your manuscript to PLOS ONE. After careful consideration, we feel that it has merit but does not fully meet PLOS ONE’s publication criteria as it currently stands. Therefore, we invite you to submit a revised version of the manuscript that addresses the points raised during the review process.

The concerns of the reviewer do not necessarily indicate that the results of the paper are incorrect, but rather that the methodology is flawed. And as mentioned by the reviewer and authors, the flaw is caused by specific events during the pandemic. We cannot avoid (ignore the flaw) or overcome it (supplement the complete control experiment). It is just like the complexity of sociological issues, which is often determined by multiple factors. The continuous contemplation and collision of ideas between the author and the reviewer in pursuit of truth in this paper have high academic contribution and reflect the diversity of sociological issues.Therefore, I suggest the author to: a) reflect the concerns of the reviewers in the discussion section. b) increase the credibility of the results as much as possible by supplementing necessary tests. c) improve the transparency of data. d) complete legends and labeling.

We look forward to receiving your revised manuscript.

Kind regards,

Bifeng Zhu

Academic Editor

PLOS ONE

Journal Requirements:

Reviewers' comments:

Reviewer's Responses to Questions

**Comments to the Author**

Reviewer #2: All comments have been addressed

Reviewer #4: (No Response)

2. Is the manuscript technically sound, and do the data support the conclusions?

Reviewer #2: Yes

Reviewer #4: No

3. Has the statistical analysis been performed appropriately and rigorously?

Reviewer #2: Yes

Reviewer #4: No

4. Have the authors made all data underlying the findings in their manuscript fully available?

Reviewer #2: Yes

Reviewer #4: No

5. Is the manuscript presented in an intelligible fashion and written in standard English?

Reviewer #2: Yes

Reviewer #4: Yes

Reviewer #2: Thank you for the attentive and comprehensive revision. The manuscript is now methodologically sound, clearly written, and ready for publication.

Reviewer #4: I would like to thank the authors for revising the manuscript and for adding a detailed rationale for the design, an expanded Limitations section, and relevant citations. I also appreciate the transparent discussion of the ethical and logistical challenges faced during the pandemic. While these additions are helpful, they do not fully address the previously mentioned concerns with the experimental design.

Referring to Paulus et al, one of the additional sources cited in the revised manuscript, a few of the conditions that must be met for a before–after study to yield valid inferences include the following:

Each participant’s baseline must accurately predict what would have happened without treatment

Changes in the environment must be addressed

The condition studied should be relatively stable (i.e. not subject to natural recovery or intermittent fluctuations).

While the authors might see some way forward to address these threats to validity, I do not. As these data were collected during the COVID-19 pandemic, a unique and sometimes rapidly changing environment, the unique context likely precludes attempts to address these issues using historical or external data. Indeed, it was implied that some outcome measures (e.g. depression scores) were likely strongly influenced by the environment.

Additionally, the highly confounded study design should effectively preclude the between-group comparative analyses reported here, rather than enabling them as the manuscript states. Prioritizing participant well-being is commendable, but it is my belief that equipoise was neglected in this study design.

I would also like to call attention to several other issues:

Lack of correction for multiple comparisons. The study conducts within-group and between-group tests across multiple metrics and timepoints without any adjustment for Type I error.

Insufficient data transparency. While the authors report summary statistics, reliance on summary statistics alone prevents readers and reviewers from evaluating the validity of the non-parametric analyses. Please provide the underlying individual-level data.

Incomplete legends and labeling. Several tables and figures lack sufficiently detailed legends or labels, making it difficult to interpret the results independently. Reported summary statistics are not clearly labeled, etc. Please ensure that all such materials are fully self-explanatory.

**Do you want your identity to be public for this peer review?** For information about this choice, including consent withdrawal, please see our Privacy Policy

Reviewer #2: No

Reviewer #4: No

---

## [Author Response · Author response to Decision Letter 3]

22 Sep 2025

Please see the attached file Response to reviewers R3.

---

## [Editor Report · Decision Letter 3]

23 Sep 2025

Aging in rural communities: Engagement in indoor leisure activities and older adult health

PONE-D-24-46685R3

Dear Dr. Chiang,

We’re pleased to inform you that your manuscript has been judged scientifically suitable for publication and will be formally accepted for publication once it meets all outstanding technical requirements.

Kind regards,

Bifeng Zhu

Academic Editor

PLOS ONE
---

## [Editor Report · Acceptance letter]

PONE-D-24-46685R3

PLOS ONE

Dear Dr. Chiang,

I'm pleased to inform you that your manuscript has been deemed suitable for publication in PLOS ONE. Congratulations! Your manuscript is now being handed over to our production team.

Kind regards,

on behalf of

Dr. Bifeng Zhu

Academic Editor

PLOS ONE